# Hardness of parameter estimation in graphical models

**Guy Bresler**[1]    **David Gamarnik**[2]    **Devavrat Shah**[1]
Laboratory for Information and Decision Systems
Department of EECS[1] and Sloan School of Management[2]
Massachusetts Institute of Technology
{gbresler,gamarnik,devavrat}@mit.edu

## Abstract

We consider the problem of learning the canonical parameters specifying an undirected graphical model (Markov random field) from the mean parameters. For graphical models representing a minimal exponential family, the canonical parameters are uniquely determined by the mean parameters, so the problem is feasible in principle. The goal of this paper is to investigate the computational feasibility of this statistical task. Our main result shows that parameter estimation is in general intractable: no algorithm can learn the canonical parameters of a generic pair-wise binary graphical model from the mean parameters in time bounded by a polynomial in the number of variables (unless RP = NP). Indeed, such a result has been believed to be true (see [1]) but no proof was known.

Our proof gives a polynomial time reduction from approximating the partition function of the hard-core model, known to be hard, to learning approximate parameters. Our reduction entails showing that the marginal polytope boundary has an inherent repulsive property, which validates an optimization procedure over the polytope that does not use any knowledge of its structure (as required by the ellipsoid method and others).

## 1   Introduction

Graphical models are a powerful framework for succinct representation of complex high-dimensional distributions. As such, they are at the core of machine learning and artificial intelligence, and are used in a variety of applied fields including finance, signal processing, communications, biology, as well as the modeling of social and other complex networks. In this paper we focus on binary pairwise undirected graphical models, a rich class of models with wide applicability. This is a parametric family of probability distributions, and for the models we consider, the canonical parameters $\theta$ are uniquely determined by the vector $\mu$ of mean parameters, which consist of the node-wise and pairwise marginals.

Two primary statistical tasks pertaining to graphical models are inference and parameter estimation. A basic inference problem is the computation of marginals (or conditional probabilities) given the model, that is, the *forward mapping* $\theta \mapsto \mu$. Conversely, the *backward mapping* $\mu \mapsto \theta$ corresponds to learning the canonical parameters from the mean parameters. The backward mapping is defined only for $\mu$ in the marginal polytope $\mathcal{M}$ of realizable mean parameters, and this is important in what follows. The backward mapping captures maximum likelihood estimation of parameters; the study of the statistical properties of maximum likelihood estimation for exponential families is a classical and important subject.

In this paper we are interested in the *computational tractability* of these statistical tasks. A basic question is whether or not these maps can be computed efficiently (namely in time polynomial in

the problem size). As far as inference goes, it is well known that approximating the forward map (inference) is computational hard in general. This was shown by Luby and Vigoda [2] for the hard-core model, a simple pairwise binary graphical model (defined in (2.1)). More recently, remarkably sharp results have been obtained, showing that computing the forward map for the hard-core model is tractable if and only if the system exhibits the correlation decay property [3, 4]. In contrast, to the best of our knowledge, no analogous hardness result exists for the backward mapping (parameter estimation), despite its seeming intractability [1].

Tangentially related hardness results have been previously obtained for the problem of learning the *graph structure* underlying an undirected graphical model. Bogdanov et al. [5] showed hardness of determining graph structure when there are hidden nodes, and Karger and Srebro [6] showed hardness of finding the maximum likelihood graph with a given treewidth. Computing the backward mapping, in comparison, requires estimation of the parameters when the graph is known.

Our main result, stated precisely in the next section, establishes hardness of approximating the backward mapping for the hard-core model. Thus, despite the problem being *statistically* feasible, it is computationally intractable.

The proof is by reduction, showing that the backward map can be used as a black box to efficiently estimate the partition function of the hard-core model. The reduction, described in Section 4, uses the variational characterization of the log-partition function as a constrained convex optimization over the marginal polytope of realizable mean parameters. The gradient of the function to be minimized is given by the backward mapping, and we use a projected gradient optimization method. Since approximating the partition function of the hard-core model is known to be computationally hard, the reduction implies hardness of approximating the backward map.

The main technical difficulty in carrying out the argument arises because the convex optimization is constrained to the marginal polytope, an intrinsically complicated object. Indeed, even determining membership (or evaluating the projection) to within a crude approximation of the polytope is NP-hard [7]. Nevertheless, we show that it is possible to do the optimization without using any knowledge of the polytope structure, as is normally required by ellipsoid, barrier, or projection methods. To this end, we prove that the polytope boundary has an inherent repulsive property that keeps the iterates inside the polytope without actually enforcing the constraint. The consequence of the boundary repulsion property is stated in Proposition 4.6 of Section 4, which is proved in Section 5.

Our reduction has a close connection to the variational approach to approximate inference [1]. There, the conjugate-dual representation of the log-partition function leads to a relaxed optimization problem defined over a tractable bound for the marginal polytope and with a simple surrogate to the entropy function. What our proof shows is that accurate approximation of the gradient of the entropy obviates the need to relax the marginal polytope.

We mention a related work of Kearns and Roughgarden [8] showing a polynomial-time reduction from inference to determining membership in the marginal polytope. Note that such a reduction does not establish hardness of parameter estimation: the empirical marginals obtained from samples are *guaranteed* to be in the marginal polytope, so an efficient algorithm could hypothetically exist for parameter estimation without contradicting the hardness of marginal polytope membership.

After completion of our manuscript, we learned that Montanari [9] has independently and simultaneously obtained similar results showing hardness of parameter estimation in graphical models from the mean parameters. His high-level approach is similar to ours, but the details differ substantially.

## 2  Main result

In order to establish hardness of learning parameters from marginals for pairwise binary graphical models, we focus on a specific instance of this class of graphical models, the hard-core model. Given a graph $G = (V, E)$ (where $V = \{1, \ldots, p\}$), the collection of independent set vectors $\mathcal{I}(G) \subseteq \{0, 1\}^V$ consist of vectors $\sigma$ such that $\sigma_i = 0$ or $\sigma_j = 0$ (or both) for every edge $\{i, j\} \in E$. Each vector $\sigma \in \mathcal{I}(G)$ is the indicator vector of an independent set. The hard-core model assigns nonzero probability only to independent set vectors, with

$$\mathsf{P}_\theta(\sigma) = \exp\left(\sum_{i \in V} \theta_i \sigma_i - \Phi(\theta)\right) \quad \text{for each} \quad \sigma \in \mathcal{I}(G). \tag{2.1}$$

This is an exponential family with vector of sufficient statistics $\phi(\sigma) = (\sigma_i)_{i \in V} \in \{0, 1\}^p$ and vector of canonical parameters $\theta = (\theta_i)_{i \in V} \in \mathbb{R}^p$. In the statistical physics literature the model is usually parameterized in terms of node-wise fugacity (or activity) $\lambda_i = e^{\theta_i}$. The log-partition function

$$\Phi(\theta) = \log \left( \sum_{\sigma \in \mathcal{I}(G)} \exp \left( \sum_{i \in V} \theta_i \sigma_i \right) \right)$$

serves to normalize the distribution; note that $\Phi(\theta)$ is finite for all $\theta \in \mathbb{R}^p$. Here and throughout, all logarithms are to the natural base.

The set $\mathcal{M}$ of realizable mean parameters plays a major role in the paper, and is defined as

$$\mathcal{M} = \{\mu \in \mathbb{R}^p \,|\, \text{there exists a } \theta \text{ such that } \mathsf{E}_\theta[\phi(\sigma)] = \mu \} \,.$$

For the hard-core model (2.1), the set $\mathcal{M}$ is a polytope equal to the convex hull of independent set vectors $\mathcal{I}(G)$ and is called the *marginal polytope*. The marginal polytope's structure can be rather complex, and one indication of this is that the number of half-space inequalities needed to represent $\mathcal{M}$ can be very large, depending on the structure of the graph $G$ underlying the model [10, 11].

The model (2.1) is a regular minimal exponential family, so for each $\mu$ in the interior $\mathcal{M}^\circ$ of the marginal polytope there corresponds a unique $\theta(\mu)$ satisfying the dual matching condition

$$\mathsf{E}_\theta[\phi(\sigma)] = \mu \,.$$

We are concerned with approximation of the backward mapping $\mu \mapsto \theta$, and we use the following notion of approximation.

**Definition 2.1.** We say that $\hat{y} \in \mathbb{R}$ is a $\delta$-approximation to $y \in \mathbb{R}$ if $y(1 - \delta) \leq \hat{y} \leq (1 + \delta)$. A vector $\hat{v} \in \mathbb{R}^p$ is a $\delta$-approximation to $v \in \mathbb{R}^p$ if each entry $\hat{v}_i$ is a $\delta$-approximation to $v_i$.

We next define the appropriate notion of efficient approximation algorithm.

**Definition 2.2.** A *fully polynomial randomized approximation scheme* (FPRAS) for a mapping $f_p : \mathcal{X}_p \to \mathbb{R}$ is a randomized algorithm that for each $\delta > 0$ and input $x \in \mathcal{X}_p$, with probability at least $3/4$ outputs a $\delta$-approximation $\hat{f}_p(x)$ to $f_p(x)$ and moreover the running time is bounded by a polynomial $Q(p, \delta^{-1})$.

Our result uses the complexity classes RP and NP, defined precisely in any complexity text (such as [12]). The class RP consists of problems solvable by efficient (randomized polynomial) algorithms, and NP consists of many seemingly difficult problems with no known efficient algorithms. It is widely believed that NP $\neq$ RP. Assuming this, our result says that there cannot be an efficient approximation algorithm for the backward mapping in the hard-core model (and thus also for the more general class of binary pairwise graphical models).

We recall that approximating the backward mapping entails taking a vector $\mu$ as input and producing an approximation of the corresponding vector of canonical parameters $\theta$ as output. It should be noted that even determining whether a given vector $\mu$ belongs to the marginal polytope $\mathcal{M}$ is known to be an NP-hard problem [7]. However, our result shows that the problem is NP-hard even if the input vector $\mu$ is known *a priori* to be an element of the marginal polytope $\mathcal{M}$.

**Theorem 2.3.** *Assuming* NP $\neq$ RP, *there does not exist an* FPRAS *for the backward mapping* $\mu \mapsto \theta$.

As discussed in the introduction, Theorem 2.3 is proved by showing that the backward mapping can be used as a black-box to efficiently estimate the partition function of the hard core model, known to be hard. This uses the variational characterization of the log-partition function as well as a projected gradient optimization method. Proving validity of the projected gradient method requires overcoming a substantial technical challenge: we show that the iterates remain within the marginal polytope without explicitly enforcing this (in particular, we do not project onto the polytope). The bulk of the paper is devoted to establishing this fact, which may be of independent interest.

In the next section we give necessary background on conjugate-duality and the variational characterization as well as review the result we will use on hardness of computing the log-partition function. The proof of Theorem 2.3 is then given in Section 4.

# 3 Background

## 3.1 Exponential families and conjugate duality

We now provide background on exponential families (as can be found in the monograph by Wainwright and Jordan [1]) specialized to the hard-core model (2.1) on a fixed graph $G = (V, E)$. General theory on conjugate duality justifying the statements of this subsection can be found in Rockafellar's book [13].

The basic relationship between the canonical and mean parameters is expressed via conjugate (or Fenchel) duality. The conjugate dual of the log-partition function $\Phi(\theta)$ is

$$\Phi^*(\mu) := \sup_{\theta \in \mathbb{R}^d} \left\{ \langle \mu, \theta \rangle - \Phi(\theta) \right\}.$$

Note that for our model $\Phi(\theta)$ is finite for all $\theta \in \mathbb{R}^p$ and furthermore the supremum is uniquely attained. On the interior $\mathcal{M}^\circ$ of the marginal polytope, $-\Phi^*$ is the entropy function. The log-partition function can then be expressed as

$$\Phi(\theta) = \sup_{\mu \in \mathcal{M}} \left\{ \langle \theta, \mu \rangle - \Phi^*(\mu) \right\}, \tag{3.1}$$

with

$$\mu(\theta) = \arg\max_{\mu \in \mathcal{M}} \left\{ \langle \theta, \mu \rangle - \Phi^*(\mu) \right\}. \tag{3.2}$$

The forward mapping $\theta \mapsto \mu$ is specified by the variational characterization (3.2) or alternatively by the gradient map $\nabla \Phi : \mathbb{R}^p \to \mathcal{M}$.

As mentioned earlier, for each $\mu$ in the interior $\mathcal{M}^\circ$ there is a unique $\theta(\mu)$ satisfying the dual matching condition $\mathsf{E}_{\theta(\mu)}[\phi(\sigma)] = (\nabla \Phi)(\theta(\mu)) = \mu$.

For mean parameters $\mu \in \mathcal{M}^\circ$, the backward mapping $\mu \mapsto \theta(\mu)$ to the canonical parameters is given by

$$\theta(\mu) = \arg\max_{\theta \in \mathbb{R}^p} \left\{ \langle \mu, \theta \rangle - \Phi(\theta) \right\}$$

or by the gradient

$$\nabla \Phi^*(\mu) = \theta(\mu).$$

The latter representation will be the more useful one for us.

## 3.2 Hardness of inference

We describe an existing result on the hardness of inference and state the corollary we will use. The result says that, subject to widely believed conjectures in computational complexity, no efficient algorithm exists for approximating the partition function of certain hard-core models. Recall that the hard-core model with fugacity $\lambda$ is given by (2.1) with $\theta_i = \ln \lambda$ for each $i \in V$.

**Theorem 3.1** ([3, 4]). *Suppose $d \geq 3$ and $\lambda > \lambda_c(d) = \frac{(d-1)^{d-1}}{(d-2)^d}$. Assuming $\mathrm{NP} \neq \mathrm{RP}$, there exists no FPRAS for computing the partition function of the hard-core model with fugacity $\lambda$ on regular graphs of degree $d$. In particular, no FPRAS exists when $\lambda = 1$ and $d \geq 5$.*

We remark that the source of hardness is the long-range dependence property of the hard-core model for $\lambda > \lambda_c(d)$. It was shown in [14] that for $\lambda < \lambda_c(d)$ the model exhibits decay of correlations and there is an FPRAS for the log-partition function (in fact there is a deterministic approximation scheme as well). We note that a number of hardness results are known for the hardcore and Ising models, including [15, 16, 3, 2, 4, 17, 18, 19]. The result stated in Theorem 3.1 suffices for our purposes.

From this section we will need only the following corollary, proved in the Appendix. The proof, standard in the literature, uses the self-reducibility of the hard-core model to express the partition function in terms of marginals computed on subgraphs.

**Corollary 3.2.** *Consider the hard-core model (2.1) on graphs of degree most $d$ with parameters $\theta_i = 0$ for all $i \in V$. Assuming $\mathrm{NP} \neq \mathrm{RP}$, there exists no FPRAS $\hat{\mu}(\mathbf{0})$ for the vector of marginal probabilities $\mu(\mathbf{0})$, where error is measured entry-wise as per Definition 2.1.*

# 4 Reduction by optimizing over the marginal polytope

In this section we describe our reduction and prove Theorem 2.3. We define polynomial constants

$$\epsilon = p^{-8}, \quad q = p^5, \quad \text{and} \quad s = \left(\frac{\epsilon}{2p}\right)^2, \tag{4.1}$$

which we will leave as $\epsilon$, $q$, and $s$ to clarify the calculations. Also, given the asymptotic nature of the results, we assume that $p$ is larger than a universal constant so that certain inequalities are satisfied.

**Proposition 4.1.** *Fix a graph $G$ on $p$ nodes. Let $\hat{\theta} : \mathcal{M}^{\circ} \rightarrow \mathbb{R}^p$ be a black box giving a $\gamma$-approximation for the backward mapping $\mu \mapsto \theta$ for the hard-core model (2.1). Using $1/\epsilon\gamma^2$ calls to $\hat{\theta}$, and computation bounded by a polynomial in $p, 1/\gamma$, it is possible to produce a $4\gamma p^{7/2}/q\epsilon^2$-approximation $\hat{\mu}(\mathbf{0})$ to the marginals $\mu(\mathbf{0})$ corresponding to all zero parameters.*

We first observe that Theorem 2.3 follows almost immediately.

*Proof of Theorem 2.3.* A standard median amplification trick (see e.g. [20]) allows to decrease the probability $1/4$ of erroneous output by a FPRAS to below $1/p\epsilon\gamma^2$ using $O(\log(p\epsilon\gamma^2))$ function calls. Thus the assumed FPRAS for the backward mapping can be made to give a $\gamma$-approximation $\hat{\theta}$ to $\theta$ on $1/\epsilon\gamma^2$ successive calls, with probability of no erroneous outputs equal to at least $3/4$. By taking $\gamma = \tilde{\gamma}q\epsilon^2 p^{-7/2}/2$ in Proposition 4.1 we get a $\tilde{\gamma}$-approximation to $\mu(\mathbf{0})$ with computation bounded by a polynomial in $p, 1/\tilde{\gamma}$. In other words, the existence of an FPRAS for the mapping $\mu \mapsto \theta$ gives an FPRAS for the marginals $\mu(\mathbf{0})$, and by Corollary 3.2 this is not possible if NP $\neq$ RP. $\square$

We now work towards proving Proposition 4.1, the goal being to estimate the vector of marginals $\mu(\mathbf{0})$ for some fixed graph $G$. The desired marginals are given by the solution to the optimization (3.2) with $\theta = \mathbf{0}$:

$$\mu(\mathbf{0}) = -\arg\min_{\mu \in \mathcal{M}} \Phi^*(\mu). \tag{4.2}$$

We know from Section 3 that for $x \in \mathcal{M}^{\circ}$ the gradient $\nabla\Phi^*(x) = \theta(x)$, that is, the backward mapping amounts to a gradient first order (gradient) oracle. A natural approach to solving the optimization problem (4.2) is to use a projected gradient method. For reasons that will be come clear later, instead of projecting onto the marginal polytope $\mathcal{M}$, we project onto the shrunken marginal polytope $\mathcal{M}_1 \subset \mathcal{M}$ defined as

$$\mathcal{M}_1 = \{\mu \in \mathcal{M} \cap [q\epsilon, \infty)^p : \mu + \epsilon \cdot e_i \in \mathcal{M} \text{ for all } i\}, \tag{4.3}$$

where $e_i$ is the $i$th standard basis vector.

As mentioned before, projecting onto $\mathcal{M}_1$ is NP-hard, and this must therefore be avoided if we are to obtain a polynomial-time reduction. Nevertheless, we temporarily assume that it is possible to do the projection and address this difficulty later. With this in mind, we propose to solve the optimization (4.2) by a projected gradient method with fixed step size $s$,

$$x^{t+1} = \mathcal{P}_{\mathcal{M}_1}(x^t - s\nabla\Phi^*(x^t)) = \mathcal{P}_{\mathcal{M}_1}(x^t - s\theta(x^t)), \tag{4.4}$$

In order for the method (4.4) to succeed a first requirement is that the optimum is inside $\mathcal{M}_1$. The following lemma is proved in the Appendix.

**Lemma 4.2.** *Consider the hard core model (2.1) on a graph $G$ with maximum degree $d$ on $p \geq 2^{d+1}$ nodes and canonical parameters $\theta = \mathbf{0}$. Then the corresponding vector of mean parameters $\mu(\mathbf{0})$ is in $\mathcal{M}_1$.*

One of the benefits of operating within $\mathcal{M}_1$ is that the gradient is bounded by a polynomial in $p$, and this will allow the optimization procedure to converge in a polynomial number of steps. The following lemma amounts to a rephrasing of Lemmas 5.3 and 5.4 in Section 5 and the proof is omitted.

**Lemma 4.3.** *We have the gradient bound $\|\nabla\Phi^*(x)\|_{\infty} = \|\theta(x)\|_{\infty} \leq p/\epsilon = p^9$ for any $x \in \mathcal{M}_1$.*

Next, we state general conditions under which an approximate projected gradient algorithm converges quickly. Better convergence rates are possible using the strong convexity of $\Phi^*$ (shown in Lemma 4.5 below), but this lemma suffices for our purposes. The proof is standard (see [21] or Theorem 3.1 in [22] for a similar statement) and is given in the Appendix for completeness.

**Lemma 4.4** (Projected gradient method). *Let $G : C \to \mathbb{R}$ be a convex function defined over a compact convex set $C$ with minimizer $x^* \in \arg\min_{x \in C} G(x)$. Suppose we have access to an approximate gradient oracle $\widehat{\nabla G}(x)$ for $x \in C$ with error bounded as $\sup_{x \in C} \|\widehat{\nabla G}(x) - \nabla G(x)\|_1 \leq \delta/2$. Let $L = \sup_{x \in C} \|\widehat{\nabla G}(x)\|$. Consider the projected gradient method $x^{t+1} = \mathcal{P}_C(x^t - s\widehat{\nabla G}(x^t))$ starting at $x^1 \in C$ and with fixed step size $s = \delta/2L^2$. After $T = 4\|x^1 - x^*\|^2 L^2/\delta^2$ iterations the average $\bar{x}^T = \frac{1}{T} \sum_{t=1}^T x^t$ satisfies $G(\bar{x}^T) - G(x^*) \leq \delta$.*

To translate accuracy in approximating the function $\Phi^*(x^*)$ to approximating $x^*$, we use the fact that $\Phi^*$ is strongly convex. The proof (in the Appendix) uses the equivalence between strong convexity of $\Phi^*$ and strong smoothness of the Fenchel dual $\Phi$, the latter being easy to check. Since we only require the implication of the lemma, we defer the definitions of strong convexity and strong smoothness to the appendix where they are used.

**Lemma 4.5.** *The function $\Phi^* : \mathcal{M}^\circ \to \mathbb{R}$ is $p^{-\frac{3}{2}}$-strongly convex. As a consequence, if $\Phi^*(x) - \Phi^*(x^*) \leq \delta$ for $x \in \mathcal{M}^\circ$ and $x^* = \arg\min_{y \in \mathcal{M}^\circ} \Phi^*(y)$, then $\|x - x^*\| \leq 2p^{\frac{3}{2}}\delta$.*

At this point all the ingredients are in place to show that the updates (4.4) rapidly approach $\mu(\mathbf{0})$, but a crucial difficulty remains to be overcome. The assumed black box $\hat{\theta}$ for approximating the mapping $\mu \mapsto \theta$ is *only defined for $\mu$ inside $\mathcal{M}$*, and thus it is not at all obvious how to evaluate the projection onto the closely related polytope $\mathcal{M}_1$. Indeed, as shown in [7], even approximate projection onto $\mathcal{M}$ is NP-hard, and no polynomial time reduction can require projecting onto $\mathcal{M}_1$ (assuming P $\neq$ NP).

The goal of the subsequent Section 5 is to prove Proposition 4.6 below, which states that the optimization procedure can be carried out without any knowledge about $\mathcal{M}$ or $\mathcal{M}_1$. Specifically, we show that thresholding coordinates suffices, that is, instead of projecting onto $\mathcal{M}_1$ we may project onto the translated non-negative orthant $[q\epsilon, \infty)^p$. Writing $\mathcal{P}_\geq$ for this projection, we show that the original projected gradient method (4.4) has *identical* iterates $x^t$ as the much simpler update rule

$$x^{t+1} = \mathcal{P}_\geq(x^t - s\theta(x^t)). \tag{4.5}$$

**Proposition 4.6.** *Choose constants as per (4.1). Suppose $x^1 \in \mathcal{M}_1$, and consider the iterates $x^{t+1} = \mathcal{P}_\geq(x^t - s\hat{\theta}(x^t))$ for $t \geq 1$, where $\hat{\theta}(x^t)$ is a $\gamma$-approximation of $\theta(x^t)$ for all $t \geq 1$. Then $x^t \in \mathcal{M}_1$, for all $t \geq 1$, and thus the iterates are the same using either $\mathcal{P}_\geq$ or $\mathcal{P}_{\mathcal{M}_1}$.*

The next section is devoted to the proof of Proposition 4.6. We now complete the reduction.

*Proof of Proposition 4.1.* We start the gradient update procedure $x^{t+1} = \mathcal{P}_\geq(x^t - s\hat{\theta}(x^t))$ at the point $x^1 = (\frac{1}{2p}, \frac{1}{2p}, \ldots, \frac{1}{2p})$, which we claim is within $\mathcal{M}_1$ for any graph $G$ for $p = |V|$ large enough. To see this, note that $(\frac{1}{p}, \frac{1}{p}, \ldots, \frac{1}{p})$ is in $\mathcal{M}$, because it is a convex combination (with weight $1/p$ each) of the independent set vectors $e_1, \ldots, e_p$. Hence $x^1 + \frac{1}{2p} \cdot e_i \in \mathcal{M}$, and additionally $x_i^1 = \frac{1}{2p} \geq q\epsilon$, for all $i$.

We establish that $x^t \in \mathcal{M}_1$ for each $t \geq 1$ by induction, having verified the base case $t = 1$ in the preceding paragraph. Let $x^t \in \mathcal{M}_1$ for some $t \geq 1$. At iteration $t$ of the update rule we make a call to the black box $\hat{\theta}(x^t)$ giving a $\gamma$-approximation to the backward mapping $\theta(x^t)$, compute $x^t - s\hat{\theta}(x^t)$, and then project onto $[q\epsilon, \infty)^p$. Proposition 4.6 ensures that $x^{t+1} \in \mathcal{M}_1$. Therefore, the update $x^{t+1} = \mathcal{P}_\geq(x^t - s\hat{\theta}(x^t))$ is the same as $x^{t+1} = \mathcal{P}_{\mathcal{M}_1}(x^t - s\hat{\theta}(x^t))$.

Now we can now apply Lemma 4.4 with $G = \Phi^*$, $C = \mathcal{M}_1$, $\delta = 2\gamma p^2/\epsilon$ and $L = \sup_{x \in C} \|\widehat{\nabla G}(x)\|_2 \leq \sqrt{p(p/\epsilon)^2} = p^{3/2}/\epsilon$. After

$$T = 4\|x^1 - x^*\|^2 L^2/\delta^2 \leq 4p(p^3/\epsilon^2)/(4\gamma^2 p^4/\epsilon^2) = 1/\gamma^2$$

iterations the average $\bar{x}^T = \frac{1}{T} \sum_{t=1}^T x^t$ satisfies $G(\bar{x}^T) - G(x^*) \leq \delta$.

Lemma 4.5 implies that $\|\bar{x}^T - x^*\|_2 \leq 2\delta p^{\frac{3}{2}}$, and since $x_i^* \geq q\epsilon$, we get the entry-wise bound $|\bar{x}_i^T - x_i^*| \leq 2\delta p^{\frac{3}{2}} x_i^*/q\epsilon$ for each $i \in V$. Hence $\bar{x}^T$ is a $4\gamma p^{7/2}/q\epsilon^2$-approximation for $x^*$. $\qquad\square$

# 5 Proof of Proposition 4.6

In Subsection 5.1 we prove estimates on the parameters $\theta$ corresponding to $\mu$ close to the boundary of $\mathcal{M}_1$, and then in Subsection 5.2 we use these estimates to show that the boundary of $\mathcal{M}_1$ has a certain repulsive property that keeps the iterates inside.

## 5.1 Bounds on gradient

We start by introducing some helpful notation. For a node $i$, let $\mathcal{N}(i) = \{j \in [p] : (i,j) \in E\}$ denote its neighbors. We partition the collection of independent set vectors as

$$\mathcal{I} = S_i \cup S_i^- \cup S_i^\varnothing \,,$$

where

$$
\begin{aligned}
S_i &= \{\sigma \in \mathcal{I} : \sigma_i = 1\} = \{\text{Ind sets containing } i\} \\
S_i^- &= \{\sigma - e_i : \sigma \in S_i\} = \{\text{Ind sets where } i \text{ can be added}\} \\
S_i^\varnothing &= \{\sigma \in \mathcal{I} : \sigma_j = 1 \text{ for some } j \in \mathcal{N}(i)\} = \{\text{Ind sets conflicting with } i\} \,.
\end{aligned}
$$

For a collection of independent set vectors $S \subseteq \mathcal{I}$ we write $\mathsf{P}(S)$ as shorthand for $\mathsf{P}_\theta(\sigma \in S)$ and

$$f(S) = \mathsf{P}(S) \cdot e^{\Phi(\theta)} = \sum_{\sigma \in S} \exp\left( \sum_{j \in V} \theta_j \sigma_j \right) .$$

We can then write the marginal at node $i$ as $\mu_i = \mathsf{P}(S_i)$, and since $S_i, S_i^-, S_i^\varnothing$ partition $\mathcal{I}$, the space of all independent sets of $G$, $1 = \mathsf{P}(S_i) + \mathsf{P}(S_i^-) + \mathsf{P}(S_i^\varnothing)$. For each $i$ let

$$\nu_i = \mathsf{P}(S_i^\varnothing) = \mathsf{P}(\text{a neighbor of } i \text{ is in } \sigma) \,.$$

The following lemma specifies a condition on $\mu_i$ and $\nu_i$ that implies a lower bound on $\theta_i$.

**Lemma 5.1.** *If $\mu_i + \nu_i \geq 1 - \delta$ and $\nu_i \leq 1 - \zeta\delta$ for $\zeta > 1$, then $\theta_i \geq \ln(\zeta - 1)$.*

*Proof.* Let $\alpha = e^{\theta_i}$, and observe that $f(S_i) = \alpha f(S_i^-)$. We want to show that $\alpha \geq \zeta - 1$.

The first condition $\mu_i + \nu_i \geq 1 - \delta$ implies that

$$
\begin{aligned}
f(S_i) + f(S_i^\varnothing) &\geq (1-\delta)(f(S_i) + f(S_i^\varnothing) + f(S_i^-)) \\
&= (1-\delta)(f(S_i) + f(S_i^\varnothing) + \alpha^{-1} f(S_i)) \,,
\end{aligned}
$$

and rearranging gives

$$f(S_i^\varnothing) + f(S_i) \geq \frac{1-\delta}{\delta} \alpha^{-1} f(S_i) \,. \tag{5.1}$$

The second condition $\nu_i \leq 1 - \zeta\delta$ reads $f(S_i^\varnothing) \leq (1 - \zeta\delta)(f(S_i) + f(S_i^\varnothing) + f(S_i^-))$ or

$$f(S_i^\varnothing) \leq \frac{1-\zeta\delta}{\zeta\delta} f(S_i)(1 + \alpha^{-1}) \tag{5.2}$$

Combining (5.1) and (5.2) and simplifying results in $\alpha \geq \zeta - 1$. $\qquad\square$

We now use the preceding lemma to show that if a coordinate is close to the boundary of the shrunken marginal polytope $\mathcal{M}_1$, then the corresponding parameter is large.

**Lemma 5.2.** *Let $r$ be a positive real number. If $\mu \in \mathcal{M}_1$ and $\mu + r\epsilon \cdot e_i \notin \mathcal{M}$, then $\theta_i \geq \ln\left(\frac{q}{r} - 1\right)$.*

*Proof.* We would like to apply Lemma 5.1 with $\zeta = q/r$ and $\delta = r\epsilon$, which requires showing that (a) $\nu_i \leq 1 - q\epsilon$ and (b) $\mu_i + \nu_i \geq 1 - r\epsilon$. To show (a), note that if $\mu \in \mathcal{M}_1$, then $\mu_i \geq q\epsilon$ by definition of $\mathcal{M}_1$. It follows that $\nu_i \leq 1 - \mu_i \leq 1 - q\epsilon$.

We now show (b). Since $\mu_i = \mathsf{P}(S_i)$, $\nu_i = \mathsf{P}(S_i^\varnothing)$, and $1 = \mathsf{P}(S_i) + \mathsf{P}(S_i^\varnothing) + P(S_i^-)$, (b) is equivalent to $\mathsf{P}(S_i^-) \leq r\epsilon$. We assume that $\mu + r\epsilon \cdot e_i \notin \mathcal{M}$ and suppose for the sake of

contradiction that $\mathsf{P}(S_i^-) > r\epsilon$. Writing $\eta_\sigma = \mathsf{P}(\sigma)$ for $\sigma \in \mathcal{I}$, so that $\mu = \sum_{\sigma \in \mathcal{I}} \eta_\sigma \cdot \sigma$, we define a new probability measure

$$\eta_\sigma' = \begin{cases} \eta_\sigma + \eta_{\sigma-e_i} & \text{if } \sigma \in S_i \\ 0 & \text{if } \sigma \in S_i^- \\ \eta_\sigma & \text{otherwise .} \end{cases}$$

One can check that $\mu' = \sum_{\sigma \in \mathcal{I}} \eta_\sigma' \sigma$ has $\mu_j' = \mu_j$ for each $i \neq j$ and $\mu_i' = \mu_i + \mathsf{P}(S_i^-) > \mu_i + r\epsilon$. The point $\mu'$, being a convex combination of independent set vectors, must be in $\mathcal{M}$, and hence so must $\mu + r\epsilon \cdot e_i$. But this contradicts the hypothesis and completes the proof of the lemma. $\qquad\square$

The proofs of the next two lemmas are similar in spirit to Lemma 8 in [23] and are proved in the Appendix. The first lemma gives an upper bound on the parameters $(\theta_i)_{i \in V}$ corresponding to an arbitrary point in $\mathcal{M}_1$.

**Lemma 5.3.** *If $\mu + \epsilon \cdot e_i \in \mathcal{M}$, then $\theta_i \leq p/\epsilon$. Hence if $\mu \in \mathcal{M}_1$, then $\theta_i \leq p/\epsilon$ for all $i$.*

The next lemma shows that if a component $\mu_i$ is not too small, the corresponding parameter $\theta_i$ is also not too negative. As before, this allows to bound from below the parameters corresponding to an arbitrary point in $\mathcal{M}_1$.

**Lemma 5.4.** *If $\mu_i \geq q\epsilon$, then $\theta_i \geq -p/q\epsilon$. Hence if $\mu \in \mathcal{M}_1$, then $\theta_i \geq -p/q\epsilon$ for all $i$.*

## 5.2 Finishing the proof of Proposition 4.6

We sketch the remainder of the proof here; full detail is given in Section D of the Supplement.

Starting with an arbitrary $x^t$ in $\mathcal{M}_1$, our goal is to show that $x^{t+1} = \mathcal{P}_{\geq}(x^t - s\hat\theta(x^t))$ remains in $\mathcal{M}_1$. The proof will then follow by induction, because our initial point $x^1$ is in $\mathcal{M}_1$ by the hypothesis.

The argument considers separately each hyperplane constraint for $\mathcal{M}$ of the form $\langle h, x \rangle \leq 1$. The distance of $x$ from the hyperplane is $1 - \langle h, x \rangle$. Now, the definition of $\mathcal{M}_1$ implies that if $x \in \mathcal{M}_1$, then $x + \epsilon \cdot e_i \in \mathcal{M}_1$ for all coordinates $i$, and thus $1 - \langle h, x \rangle \geq \epsilon \|h\|_\infty$ for all constraints. We call a constraint $\langle h, x \rangle \leq 1$ *critical* if $1 - \langle h, x \rangle < \epsilon \|h\|_\infty$, and *active* if $\epsilon \|h\|_\infty \leq 1 - \langle h, x \rangle < 2\epsilon \|h\|_\infty$. For $x^t \in \mathcal{M}_1$ there are no critical constraints, but there may be active constraints.

We first show that inactive constraints can at worst become active for the next iterate $x^{t+1}$, which requires only that the step-size is not too large relative to the magnitude of the gradient (Lemma 4.3 gives the desired bound). Then we show (using the gradient estimates from Lemmas 5.2, 5.3, and 5.4) that the active constraints have a *repulsive property* and that $x^{t+1}$ is no closer than $x^t$ to any active constraint, that is, $\langle h, x^{t+1} \rangle \leq \langle h, x^t \rangle$. The argument requires care, because the projection $\mathcal{P}_{\geq}$ may prevent coordinates $i$ from decreasing despite $x_i^t - s\hat\theta_i(x^t)$ being very negative if $x_i^t$ is already small. These arguments together show that $x^{t+1}$ remains in $\mathcal{M}_1$, completing the proof.

# 6 Discussion

This paper addresses the computational tractability of parameter estimation for the hard-core model. Our main result shows hardness of approximating the backward mapping $\mu \mapsto \theta$ to within a small polynomial factor. This is a fairly stringent form of approximation, and it would be interesting to strengthen the result to show hardness even for a weaker form of approximation. A possible goal would be to show that there exists a universal constant $c > 0$ such that approximation of the backward mapping to within a factor $1 + c$ in each coordinate is NP-hard.

### Acknowledgments

GB thanks Sahand Negahban for helpful discussions. Also we thank Andrea Montanari for sharing his unpublished manuscript [9]. This work was supported in part by NSF grants CMMI-1335155 and CNS-1161964, and by Army Research Office MURI Award W911NF-11-1-0036.

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
