[Supplementary Material]

# Supplementary Material

## A Miscellaneous proofs

### A.1 Proof of Corollary 3.2

The proof is standard and uses the self-reducibility of the hard-core model, meaning that conditioning on $\sigma_i = 0$ amounts to removing node $i$ from the graph. Fix a graph $G$ and parameters $\theta = \mathbf{0}$. We show that given an algorithm to approximately compute the marginals for induced subgraphs $H \subseteq G$, it is possible to approximate the partition function $e^{\Phi(\mathbf{0})}$, denoted here by $Z$. We first claim that

$$Z = \prod_{i=1}^{p} \frac{1}{1 - \mu_i(G \setminus [i-1])} . \tag{A.1}$$

The graph $G \setminus [i-1]$ is obtained by removing nodes labeled $1, 2, \ldots, i-1$, and $\mu_i(G \setminus [i-1])$ is the marginal at node $i$ for this graph. We use induction on the number of nodes. The base case with one node is trivial: $Z = 1 + e^0 = 2 = 1/(1 - \mu)$. Suppose now that the formula (A.1) holds for graphs on $k$ nodes and that $|V| = k + 1$. Let $Z_0$ and $Z_1$ denote the partition function summation restricted to $\sigma_1 = 0$ or $\sigma_1 = 1$, respectively. Thus

$$Z = Z_0 + Z_1 = Z_0 \left( \frac{Z_0 + Z_1}{Z_0} \right) = \frac{Z_0}{1 - \mu_1} .$$

Now $Z_0$ is the partition function of a new graph obtained by deleting vertex $i$, and the inductive assumption proves the formula.

From (A.1) we see that in order to compute a $\gamma$-approximation to $Z^{-1}$, it suffices to compute a $\gamma/p$ approximation to each of the marginals. Now for small $\gamma$, a $\gamma$ approximation to $Z^{-1}$ gives a $2\gamma$ approximation to $Z$, and this completes the proof.

### A.2 Proof of Lemma 4.2

We wish to show that $\mu(\mathbf{0}) \in \mathcal{M}_1$ for a graph $G = (V, E)$ of maximum degree $d$ and $p \geq 2^{d+1}$. Consider a particular node $i \in V$ with neighbors $N(i)$, and let $d_i = |N(i)|$ denote its degree. We use the notation $S_i, S_i^-, S_i^{\varnothing}$ defined in Subsection 5.1. A collection of independent set vectors $S \subseteq \mathcal{I}(G)$ is assigned probability $\mathsf{P}(S) = |S|/|\mathcal{I}(G)|$ for our choice $\theta = \mathbf{0}$, so it suffices to argue about cardinalities.

We first claim that $|S_i| \geq 2^{-d}|S_i^{\varnothing}|$. This follows by observing that each set in $S_i^{\varnothing}$ gets mapped to a set in $S_i$ by removing the neighbors $N_i$, and moreover at most $2^d$ sets are mapped to the same set in $S_i$. Next, we note that $|S_i| = |S_i^-|$ since the removal of node $i$ is a bijection from $S_i$ to $S_i^-$ and hence they are of the same cardinality. Combining these observations with the fact that $\mathsf{P}(S_i) + \mathsf{P}(S_i^-) + \mathsf{P}(S_i^{\varnothing}) = 1$, we get the estimate $\mu_i = \mathsf{P}(S_i) \geq 1/(2^{-d} + 2) \geq 2^{-d-1}$.

Next, we show for each coordinate $i$ that the vector $\mu' = \mu + 2^{-d-1} e_i$ is in $\mathcal{M}$, which will complete the proof that $\mu(\mathbf{0})$ is $\mathcal{M}_1$. Let $\eta_\sigma = \mathsf{P}_{\mathbf{0}}(\sigma)$ denote the probability assigned to $\sigma$ under the distribution with parameters $\theta = \mathbf{0}$, so that $\mu = \sum_{\sigma \in \mathcal{I}(G)} \eta_\sigma \cdot \sigma$. Similarly to the proof of Lemma 5.2, we define a new probability measure

$$\eta'_\sigma = \begin{cases} \eta_\sigma + 2^{-d-1} & \text{if } \sigma \in S_i \\ \eta_\sigma - 2^{-d-1} & \text{if } \sigma \in S_i^- \\ \eta_\sigma & \text{otherwise} . \end{cases}$$

This is a valid probability distribution because $\eta_\sigma \geq 2^{-d-1}$ for $\sigma \in S_i^-$. One can check that $\mu' = \sum_{\sigma \in \mathcal{I}} \eta'_\sigma \sigma$ has $\mu'_j = \mu_j$ for each $j \neq i$ and $\mu'_i = \mu_i + 2^{-d-1}$. The point $\mu'$, being a convex combination of independent set vectors, must be in $\mathcal{M}$, and hence so must $\mu + 2^{-d-1} e_i$.

# B Proofs for projected gradient method

## B.1 Proof of Lemma 4.4

The proof here is a slight modification of the proof of Theorem 3.1 in [22].

Observe first that if $\mathcal{P}$ is the projection onto a convex set, then $\mathcal{P}$ is a contraction: $\|\mathcal{P}(x) - \mathcal{P}(y)\|_2 \le \|x - y\|_2$ (cf. Prop 2.1.3 in [24]). Using the the convexity inequality $G(x) - G(x^*) \le \nabla G(x)^T (x - x^*)$, the definition $\eta = \sup_{x \in C} \|\widehat{\nabla G}(x) - \nabla G(x)\|_1$, and the update formula $x^{t+1} = x^t - s\widehat{\nabla G}(x^t)$, it follows that

$$
\begin{aligned}
G(x^t) - G(x^*) &\le \nabla G(x^t)^T (x^t - x^*) \\
&= \widehat{\nabla G}(x^t)^T (x^t - x^*) + (\widehat{\nabla G}(x^t)^T - \nabla G(x^t)^T)(x^t - x^*) \\
&\le \widehat{\nabla G}(x^t)^T (x^t - x^*) + \eta \|x^t - x^*\|_\infty \\
&= \frac{1}{s}(x^t - x^{t+1})^T (x^t - x^*) + \eta \\
&= \frac{1}{2s}(\|x^t - x^*\|_2^2 + \|x^t - x^{t+1}\|_2^2 - \|x^{t+1} - x^*\|_2^2) + \eta \\
&= \frac{1}{2s}(\|x^t - x^*\|_2^2 - \|x^{t+1} - x^*\|_2^2) + \frac{s}{2}\|\widehat{\nabla G}(x^t)\|_2^2 + \eta\,.
\end{aligned}
$$

Adding the preceding inequality for $t = 1$ to $t = T$, the sum telescopes and we get

$$
\sum_{t=1}^{T}[G(x^t) - G(x^*)] \le \frac{R^2}{2s} + \frac{s}{2}L^2 T + \eta T = RL\sqrt{T} + \eta T\,. \tag{B.1}
$$

Here we used the definitions $R = \|x^1 - x^*\|$ and $L = \sup_{x \in C} \|\widehat{\nabla G}(x)\|$ and the last equality is by the choice $s = \frac{R}{L\sqrt{T}}$. Now defining $\bar{x}^T = \frac{1}{T}\sum_{t=1}^{T} x^t$, dividing (B.1) through by $T$ and using the convexity of $G$ to apply Jensen's inequality gives

$$
G(\bar{x}^T) - G(x^*) \le \frac{RL}{\sqrt{T}} + \eta\,.
$$

Thus in order to make the right hand side smaller than $\delta$ it suffices to take $T = 4R^2 L^2 / \delta^2$ and $\eta = \delta/2$.

## B.2 Proof of Lemma 4.5

We start by showing that the gradient $\nabla \Phi$ is $p^{\frac{3}{2}}$-Lipschitz. Recall that $\nabla \Phi(\theta) = \mu(\theta)$. We prove a bound on $|\mu_i(\theta) - \mu_i(\theta')|$ by changing one coordinate of $\theta$ at a time. Let $\theta^{(r)} = (\theta_1, \ldots, \theta_r, \theta'_{r+1}, \ldots, \theta'_p)$. The triangle inequality gives

$$
|\mu_i(\theta) - \mu_i(\theta')| = \sum_{r=0}^{p-1} |\mu_i(\theta^{(r)}) - \mu_i(\theta^{(r+1)})|\,.
$$

A direct calculation shows that

$$
\frac{\partial}{\partial \theta_r} \mu_i(\theta) = \mathsf{P}(\sigma_i = \sigma_r = 1) - \mu_i(\theta)\mu_r(\theta)\,.
$$

Since this is uniformly bounded by one in absolute value, we obtain the inequality $|\mu_i(\theta) - \mu_i(\theta')| \le \|\theta - \theta'\|_1$ or

$$
\|\mu(\theta) - \mu(\theta')\|_1 \le p\|\theta - \theta'\|_1
$$

Hence

$$
\|\mu(\theta) - \mu(\theta')\|_2 \le \|\mu(\theta) - \mu(\theta')\|_1 \le p\|\theta - \theta'\|_1 \le p^{\frac{3}{2}}\|\theta - \theta'\|_2\,,
$$

i.e., $\nabla \Phi$ is $p^{\frac{3}{2}}$-Lipschitz.

Now the function $\nabla\Phi$ being $p^{\frac{3}{2}}$-Lipschitz implies that $\Phi$ is $p^{\frac{3}{2}}$-strongly smooth, where $\Phi$ is $\beta$-strongly smooth if

$$\Phi(x+\Delta) - \Phi(x) \leq \langle \nabla\Phi(x), \Delta \rangle + \frac{1}{2}\beta\|\Delta\|^2\,.$$

To see this, we write

$$\Phi(x+\Delta) - \Phi(x) = \int_0^1 \langle \nabla\Phi(x+\tau\Delta), \Delta\rangle d\tau = \langle \nabla\Phi(x), \Delta\rangle + \int_0^1 \big(\nabla\Phi(x+\tau\Delta) - \nabla\Phi(x)\big)d\tau$$

$$\leq \langle \nabla\Phi(x), \Delta\rangle + p^{\frac{3}{2}}\int_0^1 \langle\tau\Delta, \Delta\rangle d\tau$$

$$= \langle \nabla\Phi(x), \Delta\rangle + \tfrac{1}{2}p^{\frac{3}{2}}\|\Delta\|^2\,.$$

Now Theorem 6 from [25] or Chapter 5 of [26] imply that $\Phi^*$, being the Fenchel conjugate of $\Phi$, is $p^{-\frac{3}{2}}$-strongly convex, meaning

$$\Phi^*(x+\Delta) - \Phi^*(x) \geq \langle \nabla\Phi^*(x), \Delta\rangle + \tfrac{1}{2}p^{-\frac{3}{2}}\|\Delta\|^2\,.$$

This gives the desired bound on $\|x - x^*\|$ in terms of $\Phi^*(x) - \Phi^*(x^*)$.

## C    Proofs of gradient bounds

### C.1    Proof of Lemma 5.3

We suppose for the sake of deriving a contradiction that $\theta_i > p/\delta$. Let $\bar{\mu} = \mu + \delta \cdot e_i$, and let $\eta'$ be a probability measure such that $\bar{\mu} = \sum_{\sigma\in\mathcal{I}}\eta'_\sigma\sigma$. Now $\eta'(S_i) = \bar{\mu}_i \geq \delta$, and we define the non-negative measure $\gamma$ (summing to less than one) with support $S_i$ as

$$\gamma_\sigma = \begin{cases} \eta'_\sigma \cdot \frac{\delta}{\eta'(S_i)} & \text{if } \sigma \in S_i \\ 0 & \text{otherwise}\,. \end{cases}$$

In this way, $\gamma_\sigma \leq \eta'_\sigma$ and $\gamma(S_i) = \delta$. We define a new probability measure

$$\eta_\sigma = \begin{cases} \eta'_\sigma - \gamma_\sigma & \text{if } \sigma \in S_i \\ \eta'_\sigma + \gamma_{\sigma\cup\{i\}} & \text{if } \sigma \in S_i^- \\ \eta'_\sigma & \text{otherwise}\,, \end{cases} \tag{C.1}$$

and one may check that $\mu = \sum_{\sigma\in\mathcal{I}}\eta_\sigma\sigma$ and $\eta(S_i^-) \geq \gamma(S_i) = \delta$. We use the definitions in Subsection 5.1 to get

$$F_\mu(\theta) \triangleq \mu \cdot \theta - \log\Big(\sum_{\sigma\in\mathcal{I}}\exp(\sigma\cdot\theta)\Big)$$

$$= \sum_{\rho\in\mathcal{I}}\eta_\rho \log\frac{\exp(\rho\cdot\theta)}{\sum_\sigma \exp(\sigma\cdot\theta)}$$

$$\overset{(a)}{\leq} \sum_{\rho\in S_i^-}\eta_\rho \log\frac{\exp(\rho\cdot\theta)}{f(S_i^-) + e^{\theta_i}f(S_i^-) + f(S_i^\varnothing)}$$

$$\overset{(b)}{\leq} \sum_{\rho\in S_i^-}\eta_\rho \log\frac{f(S_i^-)}{e^{\theta_i}f(S_i^-)}$$

$$\leq -\eta(S_i^-)\theta_i \overset{(c)}{<} -p \overset{(d)}{\leq} -\log|\mathcal{I}| = F(\mathbf{0})\,.$$

Here (a) follows by restricting the sum to $S_i^- \subseteq \mathcal{I}(G)$ and from the fact that $\sum_\sigma \exp(\sigma\cdot\theta) = f(S_i^-) + e^{\theta_i}f(S_i^-) + f(S_i^\varnothing)$, (b) follows by retaining only the term $e^{\theta_i}f(S_i^-)$ in the denominator and replacing $\exp(\rho\cdot\theta)$ for $\rho \in S_i^-$ with $f(S_i^-) = \sum_{\rho\in S_i^-}\exp(\rho\cdot\theta)$, thereby increasing the argument to the logarithm, (c) uses the fact that $\eta(S_i^-) \geq \delta$ and the assumption that $\theta_i > p/\delta$, and (d) follows from the crude bound on number of independent sets $|\mathcal{I}| \leq 2^p$ and $\log 2 < 1$.

Finally, the relation $\theta(\mu) = \arg\max_\theta F_\mu(\theta)$ from Section 3 contradicts $F_\mu(\theta) < F(\mathbf{0})$.

## C.2 Proof of Lemma 5.4

We suppose for the sake of contradiction that $\theta_i < -p/\delta$ and show that $\theta$ cannot be the vector of canonical parameters corresponding to $\mu$.

Since $\mu \in \mathcal{M}$, there exists a non-negative measure $\eta$ so that $\mu = \sum_{\sigma \in \mathcal{I}} \eta_\sigma \sigma$, and furthermore $\eta(S_i) = \mu_i \geq \delta$. Now arguments similar to the proof of Lemma 5.3 above give

$$F_\mu(\theta) = \mu \cdot \theta - \log \left( \sum_\sigma \exp(\sigma \cdot \theta) \right)$$

$$= \sum_{\rho \in \mathcal{I}} \eta_\rho \log \frac{\exp(\rho \cdot \theta)}{\sum_\sigma \exp(\sigma \cdot \theta)}$$

$$\leq \sum_{\rho \in S_i} \eta_\rho \log \frac{\exp(\rho \cdot \theta)}{f(S_i^-) + e^{\theta_i} f(S_i^-) + f(S_i^*)}$$

$$\leq \sum_{\rho \in S_i} \eta_\rho \log \frac{e^{\theta_i} f(S_i^-)}{f(S_i^-) + e^{\theta_i} f(S_i^-) + f(S_i^*)}$$

$$\leq \sum_{\rho \in S_i} \eta_\rho \theta_i = \eta(S_i)\theta_i < -\delta p/\delta = -p \leq -\log|\mathcal{I}| = F(\mathbf{0}) \,.$$

As before, this contradicts the relation $\theta(\mu) = \arg\max_\theta F_\mu(\theta)$.

# D    Proof of Proposition 4.6

Starting with $x^t$ in $\mathcal{M}_1$, our goal is to show that $x^{t+1} = \mathcal{P}_\geq(x^t - s\hat{\theta}(x^t))$ remains in $\mathcal{M}_1$. The proof will then follow by induction, because our initial point $x^1$ is in $\mathcal{M}_1$ by the hypothesis.

We will use the fact that all hyperplane constraints for $\mathcal{M}$, except for the non-negativity constraints $x_i \geq 0$, can be written as $\langle h, x \rangle \leq 1$ for a vector $h \in [0,1]^p$. This can be justified using the fact that $e_i \in \mathcal{M}$ for each $i$ together with the property that for any $\mu \in \mathcal{M}$, any coordinate of $\mu$ can be set to zero while remaining in $\mathcal{M}$.

Given our current iterate $x^t$, we call a constraint $\langle h, x \rangle \leq 1$ *active* if

$$1 - 2\epsilon\|h\|_\infty < \langle h, x^t \rangle \leq 1 - \epsilon\|h\|_\infty \tag{D.1}$$

and *critical* if

$$1 - \epsilon\|h\|_\infty < \langle h, x^t \rangle \,. \tag{D.2}$$

Observe that an active constraint has a coordinate $i$ (namely $i$ with $h_i = \|h\|_\infty$) with $\langle h, x^t + 2\epsilon \cdot e_i \rangle = \langle h, x^t \rangle + 2h_i\epsilon > 1$ and similarly a critical constraint has a coordinate $i$ with $\langle h, x^t + \epsilon \cdot e_i \rangle = \langle h, x^t \rangle + h_i\epsilon > 1$.

For $x^t \in \mathcal{M}_1$ there are (by definition) no critical constraints, but there may be active constraints. We will first show that inactive constraints can at worst become active for the next iterate $x^{t+1}$, which requires only that the step-size is not too large relative to the magnitude of the gradient. Then we show that the active constraints have a repulsive property and that $x^{t+1}$ is no closer than $x^t$ to any active constraint, that is, $\langle h, x^{t+1} \rangle \leq \langle h, x^t \rangle$. Thus, if $x^t$ is in $\mathcal{M}_1$, then there are no critical constraints for $x^{t+1}$ and every coordinate $i$ satisfies $\langle h, x^{t+1} + \epsilon \cdot e_i \rangle \leq 1$ for all constraint vectors $h$. Since the projection $\mathcal{P}_\geq$ ensures that $x_i^{t+1} \geq q\epsilon$, the update $x^{t+1}$ is in $\mathcal{M}_1$. We now focus on inactive constraints.

**Inactive constraint.**    We consider an inactive constraint $h$, meaning that $\langle h, x^t \rangle + 2\epsilon\|h\|_\infty \leq 1$. By assumption the step size $s = \left(\frac{\epsilon}{2p}\right)^2$ so the increment in any coordinate $j$ is bounded as

$$x_j^{t+1} - x_j^t \leq s|\hat{\theta}_j(x^t)|$$

$$\leq s|\hat{\theta}_j(x^t) - \theta_j(x^t)| + s|\theta_j(x^t)|$$

$$\leq (1+\gamma)s|\theta_j(x^t)|$$

$$\leq \epsilon/p$$

using Lemma 5.3 and fact that $\gamma \leq 1$. These bounds give

$$\langle h, x^{t+1}\rangle = \langle h, x^t\rangle + \langle h, x^{t+1} - x^t\rangle \leq \langle h, x^t\rangle + \sum_j h_j(x_j^{t+1} - x_j^t)$$

$$\leq \langle h, x^t\rangle + p(\epsilon/p)\|h\|_\infty \leq 1 - \epsilon\|h\|_\infty$$

which shows that the constraint is not critical for $x^{t+1}$ and at worst becomes active.

**Active constraint.** The rough idea is that if a coordinate $i$ cannot be increased by $2\epsilon$ while remaining in $\mathcal{M}$, then the parameter $\theta_i$ must be sufficiently large, and the next iterate $x^{t+1}$ will decrease enough to overcome the possible increase in other coordinates. This argument does not work, however, because it might be the case that $x_i^t = q\epsilon$, which prevents any decrease (i.e., $x_i^{t+1} \geq x_i^t$) due to the projection $\mathcal{P}_\geq$. Instead, we start by showing that if *some* coordinate cannot be increased by $2\epsilon$, then there must be a *reasonably large* coordinate which cannot be increased by $4p\epsilon$.

**Lemma D.1.** *If $h$ is an active constraint, then there is a coordinate $\ell \in V$ with $x^t + (4p\epsilon)e_\ell \notin \mathcal{M}$ and $x_\ell^t \geq 2q\epsilon$.*

*Proof.* If $h$ is active then $1 - 2\epsilon\|h\|_\infty < \langle h, x^t\rangle$. Using the fact that $h_j \leq 1$ for all $j$ we have

$$1 - 2\epsilon \leq 1 - 2\epsilon\|h\|_\infty < \langle h, x^t\rangle. \tag{D.3}$$

Let $B \subseteq V$ consist of coordinates $j$ with small entries $x_j^t \leq 2\epsilon q$. Then

$$\langle h, x^t\rangle = \sum_{j\in B} h_j x^t + \sum_{j\in B^c} h_j x^t \leq |B|(2\epsilon q) + \sum_{j\in B^c} h_j x^t \leq \frac{2}{p} + \sum_{j\in B^c} h_j x_j^t. \tag{D.4}$$

The last inequality used the crude estimate $|B| \leq p$. Combining (D.3) and (D.4) and rearranging gives

$$\sum_{j\in B^c} h_j x_j^t \geq 1 - 2\epsilon - 2/p \geq 1 - 3/p,$$

and it follows that there is an $\ell \in B^c$ for which $h_\ell \geq h_\ell x_\ell^t \geq 1/2p$. Adding $h_\ell \cdot (4p\epsilon) \geq 2\epsilon$ to both sides of (D.3) shows that $x^t + (4p\epsilon)e_\ell$ violates the inequality $\langle h, x\rangle \leq 1$. This proves the lemma, since $x_\ell^t > 2q\epsilon$ for $\ell \in B^c$. □

We are now ready to prove that $\langle h, x^{t+1}\rangle \leq \langle h, x^t\rangle$. Let $\ell$ be the coordinate promised by Lemma D.1, with $x^t + (4p\epsilon)e_\ell \notin \mathcal{M}$ and $x_\ell^t \geq 2q\epsilon$. From Lemma 5.2, we know that $\theta_\ell(x^t) \geq \log\left(\frac{q}{4p} - 1\right) \geq 3\log p$, for $p$ large enough. By definition of $\hat\theta$ being a $\gamma$-approximation to $\theta$, $\hat\theta_\ell(x^t) \geq (1-\gamma)\theta_\ell(x^t)$. Therefore, since $\gamma \to 0$ as $p \to \infty$, it follows that for $p$ large enough $\hat\theta_\ell(x^t) \geq \log p$. This implies

$$x_\ell^{t+1} - x_\ell^t \leq -\min(s\hat\theta(x^t), s\log p) \leq -s\log p. \tag{D.5}$$

Here we used the fact that $x_\ell^t \geq q\epsilon + s\log p$ so the projection $\mathcal{P}_\geq$ does not affect this coordinate.

Denote by $D$ the set of coordinates

$$D = \{j \in [p] : \langle h, x^t\rangle + \tfrac{q}{2}\epsilon h_j > 1\}.$$

These coordinates have non-positive increment: since $x_j \geq q\epsilon$ for $x \in \mathcal{M}_1$, Lemma 5.1 implies that $\theta_j \geq 0$, and hence $\hat\theta_j \geq (1-\gamma)\theta_j \geq 0$, or

$$x_j^{t+1} - x_j^t \leq 0 \quad \text{for } j \in D.$$

In contrast, coordinates in $D^c$ might increase, but by a limited amount: since $x^t \in \mathcal{M}_1$, all coordinates $j \in D^c$ satisfy $x_j^t \geq q\epsilon$, and Lemma 5.4 gives the bound $\theta_j \geq -p/q\epsilon$, or

$$x_j^{t+1} - x_j^t \leq (1+\gamma)| - s\theta_j| \leq 2sp/q\epsilon \quad \text{for all } j \in D^c. \tag{D.6}$$

Additionally, by the definition of $D$ and the fact that increasing coordinate $\ell$ by $4p\epsilon$ violates $\langle h, x\rangle \leq 1$, if $j \in D^c$, then $4p\epsilon h_\ell > q\epsilon h_j/2$, or

$$h_j < 8ph_\ell/q \quad \text{for all } j \in D^c. \tag{D.7}$$

Using the crude bound $|D^c| \leq p$ together with (D.6) and (D.7) gives

$$\sum_{j \in D^c} h_j(x_j^{t+1} - x_j^t) \leq |D^c| \frac{8ph_\ell}{q} \cdot \frac{2sp}{q\epsilon} \leq s\frac{4p^2}{q^2\epsilon}h_\ell \leq 4sh_\ell. \tag{D.8}$$

Counting the contributions from $D^c$ in (D.8) in addition to $D$ (none) and $\ell$ (negative as per (D.5)), it follows that

$$\begin{aligned}
\langle c, x^{t+1} \rangle = \langle h, x^t \rangle + \langle h, x^{t+1} - x^t \rangle &\leq \langle h, x^t \rangle + sh_\ell(4 - \theta_\ell) \\
&\leq \langle h, x^t \rangle + sh_\ell(4 - \ln p) \\
&\leq \langle h, x^t \rangle.
\end{aligned}$$

Here we have used the fact that $p$ is large enough ($p \geq e^4$ suffices for this last step). In words, we move away from any active hyperplane constraint. This completes the proof.