[Reviews · NeurIPS 2014]

Submitted by Assigned_Reviewer_1

This paper studies the worst case hardness of estimating the parameters of a
binary pairwise undirected graphical model. By considering the specific case of
the hard-core / independent set model and relying on the known result that
approximating the partition function for this problem given the parameters (even
for the "unweighted" case of all theta_i being 0) is hard, the authors show that
the other direction -- namely, approximating the parameters given the node
marginals -- is hard, in the sense that it does not admit an FPRAS.

This is a strong theoretical paper, addressing a computational complexity
problem that occurs very often in theory and practice, has been conjectured to
be hard, but hadn't yet been shown formally to be hard (to the best of my and
the authors' knowledge). The work is unlikely to have much impact in practice.
Nonetheless, it provides new insights into the parameter estimation problem
through the structure of their reduction, which essentially says that if one
could efficiently approximate the parameters given the marginals, then one could
use the projected gradient method to efficiently estimate the marginals given
the parameters (in particular, when theta_i = 0), which in turn is known to be
hard.

The technique used in the proof, specifically that it suffices to actively
project on to a quadrant and yet indirectly achieve projection onto the shrunken
polytope M_1, is the main novelty. The overall reasoning seems correct: I could
not identify any significant flaws, however I have not verified the details of
all proofs, especially those in the supplementary material.

The paper is written well and often provides abstract summaries of the technical
statements/claims to follow. One aspect that could improve the paper (or perhaps
is already there but I missed) is a clear intuitive description of the
reduction, along the lines that one can efficiently approximate the
hard-to-approximate \theta -> \mu problem (specifically for \theta = 0) if one
had an efficient approximation to the \mu -> \theta problem, using an
optimization method that is guaranteed to involve only a polynomial number of
steps.

Minor comments:

- Depending on the background of the readers, they may be very familiar with one
of 'hard-code model' and 'independent set' problems, but likely not with both.
Clarifying early on that both are the same (where for the independent set
problem one associates a weight lambda_i with each node, which then induces a
weight on every independent set) would help such readers.

- In Theorem 3.1, did you mean to write lambda = 1 and d >= 6 (rather than 5)?
Otherwise it seems that lambda will be less than (d-1)^{d-1} / (d-2)^d.

- In several places, inline math expressions of the form abc/def are ambiguous.
I believe in most places you mean abc/(def) rather than (abc/d) * ef. It would
be better to remove than ambiguity.

- Page 3, in the top-third: in what sense is the marginal polytop M a polytope
"equal to the convex hull of independent set vectors I(G)"?

- Definition 2.1: missing "y" after (1+delta).

- Corollary 3.2: "degree most d" -> "degree at most d"

Summary: A strong theoretical paper showing that parameter estimation is hard (specifically, no FPRAS) for binary pairwise MRFs. Uses some delicate arguments to show that the reduction works in poly time.

Submitted by Assigned_Reviewer_18

This paper considers the computational tractability of parameter estimation for the hard-core MRF model. The main result (Theorem 2.3) in the paper is the hardness of approximating the backward mapping from mean parameters to canonical parameters.
The result, to the best of my knowledge, is new.
However, there are two questions I would like to ask before I can judge the quality of the paper.

First, what's the dimensionality of the parameter space? Is it p? How many nodes are there in the graph V? Is it p? I got confused because these two numbers are usually different.

Second, I have been using contrastive divergence as parameter estimators for MRFs. It uses gradient ascent to find MLE of the paramters. It usually runs pretty fast and yields satisfactory results. Is it somehow related to the results in your paper?

Minors:

Usually a random vector should be bold faced

In the first formula in Section 3.1, the dimension # should be p rather than d

On Line 99, {0,1}^V should be {0,1}^|V|
Summary: A good paper, but some parts are unclear.

Submitted by Assigned_Reviewer_30

This paper shows that the backward mapping from mean parameters to canonical parameters is hard for the hard-core problems. The hardness is proved by showing no FPRAS exists for such problem.

Although the paper is reasonably well-written, I feel that there are a few major drawbacks.
First of all, the main theorems of the paper only apply on the "hard-core model", a model which is not commonly used in machine learning. The title of the paper should be more specific, e.g. "Hardness of parameter estimation in Hard-core models".
Secondly, the result that backward mapping is hard in graphical model is not surprising at all. If the backward mapping is easy, then maximum likelihood is trivial to estimate because it is essentially a moment matching problem. Also, due to convex conjugacy, the hardness of forward mapping is an indication of the hardness of backward mapping.
Therefore, I think that the main interest in graphical model is on finding a small subset of tractable useful models or good approximation strategies, instead of proving its estimation is a hard problem in a specific class of models.

Minor edits:
- Definition 2.1, missing y in (1+\delta).

After author feedback:
-- The results would be much stronger, if the similar methodology applies on the Ising model. I strongly suggest that the author adds this in the next revision.
Summary: The hardness proof is only based on a specific graphical model. The motivation of the work is a bit weak.
Author Feedback
Author rebuttal: We thank the reviewers for their comments and suggestions.

We briefly reiterate the motivation of the paper, which is relevant to the reviews.

Parameter estimation is an important step in practical application of graphical models. How best to estimate parameters is an area of active research and numerous papers on the topic have appeared in NIPS over the years. Given the importance of this inherently computational problem, it is of interest to understand its computational complexity. Intriguingly, it is not known whether parameter estimation is tractable for general graphical models. We address this problem by showing that parameter estimation is intractable for the hard-core model from statistical physics. The hard-core model is an example pairwise binary graphical model and thus intractability for the hard-core model shows intractability for this larger class of binary graphical models.

The question of tractability of parameter estimation in graphical models is a known open problem. It has been loosely conjectured to be intractable in various places over the years, including in the 2008 monograph on graphical models by Wainwright and Jordan, but until the present paper a proof of this fact was not known. The variational approach to inference is based on using Fenchel duality (convex conjugacy) to relate parameter estimation (backward mapping) to computing marginals (forward mapping), and our proof uses this as a starting point. However, the result requires going significantly beyond this starting point: as part of the proof we show intricate properties of the marginal polytope related to a certain repelling property which we introduce and study in the paper.

The main point raised by Reviewer 1:
1) The reviewer offered the following suggestion: “One aspect that could improve the paper (or perhaps is already there but I missed) is a clear intuitive description of the reduction, along the lines that one can efficiently approximate the
 hard-to-approximate \theta -> \mu problem (specifically for \theta = 0) if one 
had an efficient approximation to the \mu -> \theta problem, using an
 optimization method that is guaranteed to involve only a polynomial number of 
steps.”
Response: In the revision of the paper we will be happy to provide such a summary of the steps of our analysis.

The main points raised by Reviewer 2:
1) What is p? Both graph size and dimension of parameter space?
Response: p is the number of parameters as well as the number of nodes in the graph (as described in Section 2 near the bottom of page 2). For the hard-core (independent set) model these two numbers are the same, with a single parameter per node (known as activity or fugacity in the statistical physics community). The parameters corresponding to pairwise interactions are all equal to zeros and minus infinity.
2) This reviewer uses a gradient ascent method to efficiently and accurately learn parameters of some graphical models. How do these observations relate to the present paper?
Response: Our paper does not preclude estimating parameters for specific model subclasses, and thus our results do not contradict the reviewer’s observation that for some problems in practice, parameters can be estimated efficiently. Indeed, in light of our general hardness result, it is of interest to identify families of models for which parameter estimation is provably feasible. But as our main result shows, efficient parameter estimation is not possible for all pairwise binary models.

The main points raised by Reviewer 3:
1) The results apply only to the hard-core model.
Response: We have two explanations in this regard. First, the goal of the paper is to show intractability of parameter estimation in general binary pairwise graphical models, and showing intractability for a specific model subclass suffices for this purpose.
Second, we agree that our results as stated do not preclude tractable learning of parameters for (say) Ising models (which have edge parameters and are used extensively in machine learning). However, using a similar approach to that in the paper we are able to show analogous results for hardness of parameter estimation in Ising models. The derivations are lengthier and will not fit within the page limit, so we opted to focus on the hard-core model; we will add a note in the paper describing how the results apply more generally.
2) The second concern is that the result is “not surprising at all”, and somehow ought to follow directly from Fenchel duality (convex conjugacy).
Response: We are not aware of any direct simple proof based on duality. The hardness of parameter estimation has been loosely conjectured before, but a proof was not known: This is our contribution. The proof is non-obvious and requires a delicate argument going significantly beyond a straightforward application of Fenchel duality. Whether the result is surprising is a subjective matter, and in any case requires a rigorous proof.